# Cadmium-Induced Tubular Dysfunction in Type 2 Diabetes: A Population-Based Cross-Sectional Study

**DOI:** 10.3390/toxics11040390

**Published:** 2023-04-21

**Authors:** Soisungwan Satarug, Supabhorn Yimthiang, Phisit Pouyfung, Tanaporn Khamphaya, David A. Vesey

**Affiliations:** 1The Centre for Kidney Disease Research, Translational Research Institute, Brisbane 4102, Australia; 2Occupational Health and Safety, School of Public Health, Walailak University, Nakhon Si Thammarat 80160, Thailand; 3Department of Kidney and Transplant Services, Princess Alexandra Hospital, Brisbane 4102, Australia

**Keywords:** β_2_-microglobulin, cadmium, diabetes, diabetic nephropathy, GFR, tubular proteinuria

## Abstract

The global prevalence of diabetes, and its major complication, diabetic nephropathy, have reached epidemic proportions. The toxic metal cadmium (Cd) also induces nephropathy, indicated by a sustained reduction in the estimated glomerular filtration rate (eGFR) and the excretion of β_2_-microglobulin (β_2_M) above 300 µg/day, which reflects kidney tubular dysfunction. However, little is known about the nephrotoxicity of Cd in the diabetic population. Here, we compared Cd exposure, eGFR, and tubular dysfunction in both diabetics (*n* = 81) and non-diabetics (*n* = 593) who were residents in low- and high-Cd exposure areas of Thailand. We normalized the Cd and β_2_M excretion rates (E_Cd_ and E_β2M_) to creatinine clearance (C_cr_) as E_Cd_/C_cr_ and E_β2M_/C_cr_. Tubular dysfunction and a reduced eGFR were, respectively, 8.7-fold (*p* < 0.001) and 3-fold (*p* = 0.012) more prevalent in the diabetic than the non-diabetic groups. The doubling of E_Cd_/C_cr_ increased the prevalence odds ratios for a reduced eGFR and tubular dysfunction by 50% (*p* < 0.001) and 15% (*p* = 0.002), respectively. In a regression model analysis of diabetics from the low-exposure locality, E_β2M_/C_cr_ was associated with E_Cd_/C_cr_ (β = 0.375, *p* = 0.001) and obesity (β = 0.273, *p* = 0.015). In the non-diabetic group, E_β2M_/C_cr_ was associated with age (β = 0.458, *p* < 0.001) and E_Cd_/C_cr_ (β = 0.269, *p* < 0.001). However, after adjustment for age, and body mass index (BMI), E_β2M_/C_cr_ was higher in the diabetics than non-diabetics of similar E_Cd_/C_cr_ ranges. Thus, tubular dysfunction was more severe in diabetics than non-diabetics of similar age, BMI, and Cd body burden.

## 1. Introduction

Type 2 diabetes is a metabolic disorder resulting in a rise of fasting plasma glucose ≥ 126 mg/dL. The worldwide prevalence of diabetes, which is often linked to obesity, has now reached epidemic proportions. However, it is increasingly apparent that exposure to various diabetogenic pollutants, such as cadmium (Cd), is an important risk factor [1,2,3,4,5,6]. Strong evidence supporting the diabetogenicity of Cd comes from the Wuhan–Zhuhai prospective cohort study. This study, which measured the fasting plasma glucose levels and urinary Cd over a 3-year period, reported that for each tenfold increase in urinary Cd, the prevalence of prediabetes increased by 42% [4].

Although Cd exposure has only recently been recognized as a risk factor for diabetes [5,6], the increased susceptibility to Cd-induced nephrotoxicity in people with diabetes was first noted in the 1990 Belgian population study (Cadmibel) [7]. Similar observations were then made in studies from Sweden [8,9], Australia [10], the U.S. [11], and Korea [12]. The defective tubular reabsorption of proteins, indicated by an increase in the excretion of proteins of low molecular weight, such as β_2_-microglobulin (β_2_M), is the most frequently reported sign of Cd nephrotoxicity [13,14]. In early studies, the effect of Cd on the glomerular filtration rate (GFR) involved workplace exposure, where workers were exposed mainly through the inhalation of relatively high levels of Cd in fumes and dust [15,16]. Evidence that low environmental Cd exposure through the diet also affects the GFR has now been reported in adult populations in many countries, including the U.S. [11,17,18,19], Thailand [20], Guatemala [21], Myanmar [22], Taiwan [23], and Korea [24,25].

Epidemiologic studies implicating low environmental exposure to Cd in the pathogenesis of tubulopathy and GFR deterioration are abundant. In comparison, studies on these Cd-induced pathologies in the diabetic population are limited. Therefore, the present study aimed to quantify Cd exposure levels, kidney tubular dysfunction, and the reduction of eGFR experienced by diabetics and non-diabetics who live in low- and high-Cd-exposure areas of Thailand.

We used the excretion of Cd as a measure of long-term exposure (body burden) and a sign of nephrotoxicity. The utility of Cd excretion as an indicator of its toxicity to kidneys is based on a study showing that Cd excreted in complexes with metallothionein (MT) emanated from injured or dying kidney tubular epithelial cells [26,27]. Tubular proteinuria was indicated by a rise of β_2_M excretion ≥ 300 µg/L of filtrate. We employed established equations of the Chronic Kidney Disease Epidemiology Collaboration (CKD-EPI) to compute the estimated GFR (eGFR) [28].

For the accurate quantification of the kidney burden of Cd and its effects, we normalized the excretion of Cd and β_2_M (E_Cd_ and E_β2M_) to creatinine clearance (C_cr_), denoted as E_Cd_/C_cr_ and E_β2M_/C_cr_, respectively [29]. This C_cr_ normalization depicts an amount of a given chemical excreted per volume of filtrate, which is at least roughly related to the amount of the chemical excreted per nephron. In effect, C_cr_ normalization corrects for differences in the number of surviving nephrons among study subjects [29].

## 2. Materials and Methods

### 2.1. Cohort Participants

Participants came from three population-based studies undertaken in a Cd pollution area in the Mae Sot District, Tak Province (*n* = 211), two low-Cd-exposure locations in Bangkok (*n* = 322), and the Pakpoon municipality of Nakhon Si Thammarat Province (*n* = 141). The Institutional Ethical Committees of Chulalongkorn University, Chiang Mai University, and the Mae Sot Hospital approved the study protocol for the Mae Sot and Bangkok groups [30]. The Office of the Human Research Ethics Committee of Walailak University in Thailand approved the study protocol for the Pakpoon group [31]. All participants gave informed consent, and all had resided at their current addresses for at least 30 years. Exclusion criteria included pregnancy, breastfeeding, a history of metal work, and a hospital record or physician’s diagnosis of an advanced chronic disease. Smoking, diabetes, hypertension, regular use of medications, educational level, occupation, and family health history were ascertained by questionnaire. Prediabetes and diabetes were indicated by fasting plasma glucose levels ≥110 and ≥126 mg/dL, respectively (https://www.cdc.gov/diabetes/basics/getting-tested.html) (accessed on 17 February 2023) or a physician’s prescription of anti-diabetic medications. Hypertension was defined as systolic blood pressure ≥ 140 mmHg, diastolic blood pressure ≥ 90 mmHg, a physician’s diagnosis, or prescription of anti-hypertensive medications.

As workplace exposure was an exclusion criterion, participants presumably acquired Cd from the diet and/or smoking. Based on the measured levels of Cd in duplicate diets [32] and a nationwide survey of Cd levels in soils and food crops [33], environmental exposure to Cd in Bangkok and Nakhon Si Thammarat were low, and the Mae Sot district was high. From a previous study conducted in the Mae Sot district, the Cd content of the paddy soil samples exceeded the standard of 0.15 mg/kg, and rice samples collected from households contained 4 times the amount of the permissible Cd level of 0.1 mg/kg [34]. A health survey reported that the prevalence of CKD in the Mae Sot was 16.1%, while the prevalence of tubular proteinuria was 36.1% [35].

### 2.2. Urine and Blood Sampling and Analysis

Second morning urine samples were collected after an overnight fast. Samples of whole blood were obtained within 3 h of urine sampling. Aliquots of urine, whole blood, and plasma were stored at −20 °C or −80 °C for later analysis. The assay for the urine and plasma concentrations of creatinine ([cr]_u_ and [cr]_p_) was based on the Jaffe reaction. The assay of β_2_M in the urine ([β_2_M]_u_) was based on the latex immunoagglutination method (LX test, Eiken 2MGII; Eiken and Shionogi Co., Tokyo, Japan).

For the Bangkok group, the urinary concentration of Cd ([Cd]_u_) was determined by inductively coupled plasma mass spectrometry (ICP/MS, Agilent 7500, Agilent Technologies, Santa Clara, CA, USA). Multi-element standards (EM Science, EM Industries, Inc., Newark, NJ, USA) were used to calibrate the Cd analyses. The accuracy and precision of those analyses were ascertained with reference urine (Lyphochek^®^, Bio-Rad, Sydney, Australia). The [Cd]_u_ assigned to samples with Cd below the detection limit was 0.05 µg/L divided by the square root of 2 [36].

For the Pakpoon group, the [Cd]u was determined with the GBC System 5000 graphite furnace atomic absorption spectrophotometer (AAS) (GBC Scientific Equipment, Hampshire, IL, USA). Multielement standards were used to calibrate the metal analysis (Merck KGaA, Darmstadt, Germany). Reference urine levels 1, 2, and 3 (Lyphocheck, Bio-Rad, Hercules, CA, USA) were used for quality control, analytical accuracy, and precision assurance. When a [Cd]_u_ level was less than its detection limit, the concentration assigned was 0.1 µg/L divided by the square root of 2 [36].

For the Mae Sot group, the [Cd]u was determined with AAS (Shimadzu Model AA-6300, Kyoto, Japan). Urine standard reference material No. 2670 (National Institute of Standards, Washington, DC, USA) was used for quality assurance and control purposes. The limit of detection of the [Cd]_u_ was 0.06 µg/L. None of the urine samples contained a [Cd]_u_ below the detection limit.

The comparability of the [Cd]_u_ was ascertained by simultaneous quantification of Cd in the reference urine samples where the coefficient of variation was within acceptable clinical chemistry standards.

### 2.3. Normalization of ECd to Ccr and Ecr

E_x_ was normalized to C_cr_ as E_x_/C_cr_ = [x]_u_[cr]_p_/[cr]_u_, where x = Cd or β_2_M; [x]_u_ = urine concentration of x (mass/volume); [cr]_p_ = plasma creatinine concentration (mg/dL); and [cr]_u_ = urine creatinine concentration (mg/dL). E_x_/C_cr_ was expressed as an amount of x excreted per volume of filtrate [29].

E_x_ was normalized to E_cr_ as [x]_u_/[cr]_u_, where x = Cd or β_2_M; [x]_u_ = urine concentration of x (mass/volume); and [cr]_u_ = urine creatinine concentration (mg/dL). The ratio [x]_u_/[cr]_u_ was expressed in μg/g of creatinine.

### 2.4. Estimated Glomerular Filtration Rate

The GFR is the product of the nephron number and the mean single nephron GFR, and, in theory, the GFR is indicative of nephron function [28,37]. In practice, the GFR is estimated from established CKD-EPI equations and reported as eGFR [28].

The male eGFR = 141 × [plasma creatinine/0.9]^Y^ × 0.993 age, where Y = −0.411 if [cr]_p_ ≤ 0.9 mg/dL and Y = −1.209 if [cr]_p_ > 0.9 mg/dL. The female eGFR = 144 × [plasma creatinine/0.7]^Y^ × 0.993^age^, where Y = −0.329 if [cr]_p_ ≤ 0.7 mg/dL and Y = −1.209 if [cr]_p_ > 0.7 mg/dL.

### 2.5. Statistical Analysis

The data were analyzed with IBM SPSS Statistics 21 (IBM Inc., New York, NY, USA). We used the Mann–Whitney U-test to assess differences in the means between the two groups. The Pearson chi-squared test was used to assess differences in the percentages. To identify the departures of continuous variables from a normal distribution, the one-sample Kolmogorov–Smirnov test was used, and logarithmic transformation was applied to variables that showed rightward skewing before they were subjected to parametric statistical analysis. Logistic regression analysis was used to determine the prevalence odds ratio (POR) for tubular proteinuria and reduced eGFR. Tubular proteinuria was defined as E_β2M_/C_cr_ ≥ 300 µg/L of filtrate. Reduced eGFR was assigned when eGFR ≤ 60 mL/min/1.73 m^2^. Univariate/covariance analyses with Bonferroni correction in multiple comparisons were used to obtain the mean E_β2M_/C_cr_ adjusted for age and BMI, and interactions among groups of diabetics and non-diabetics stratified by three ranges of E_Cd_/C_cr,_. For all tests, *p*-values ≤ 0.05 were considered to indicate statistical significance.

## 3. Results

### 3.1. Cohort Composition and Characteristics

The Thai cohort of 674 participants consisted of 463 drawn from two low-exposure localities, and 211 from an area where environmental Cd pollution is endemic (Table 1).

Females constituted 57.6% of the cohort, and they formed 74.1% of the diabetic group (*n* = 81). Seventy subjects with diagnosed diabetes (86.4%) were residents of the low-exposure location, Pakpoon, and 11 diabetic cases were from a high-exposure area. The diabetic group was older (58.5 years) than the non-diabetic group (45.6 years), and the overall mean age was 47.2 years. The percentage (%) of smoking was lower in diabetics (14.9%) than in non-diabetics (34.2%). The % of obese participants was higher in the diabetics than in the non-diabetics (14.8% vs. 2.4%).

The diabetic group had a lower mean eGFR and a higher % of reduced eGFR, defined as eGFR ≤ 60 mL/min/1.73 m^2^, compared to non-diabetics (22.2% vs. 9.9%). The mean E_Cd_/C_cr_ and mean E_Cd_/E_cr_ in diabetics were all lower than in non-diabetics. Conversely, the mean E_β2M_/C_cr_ and mean E_β2M_/E_cr_ in diabetics were all higher than in non-diabetics. Nearly half (49.4%) of the diabetic group and 14.8% of the non-diabetic group had severe tubular proteinuria (E_β2M_/C_cr_ values ≥ 1000 µg/L filtrate).

### 3.2. Predictors of Tubular Proteinuria and Reduced eGFR

We employed logistic regression analysis to screen factors that may increase the risk of tubular proteinuria and/or reduced eGFR. In this analysis, the independent variables were age, BMI, log_2_[(E_Cd_/C_cr_) × 10^5^], hypertension, smoking, gender, and diabetes (Table 2).

BMI, hypertension, smoking, and gender did not show significant associations with tubular dysfunction or a reduced eGFR. Three other independent variables, namely, age, diabetes, and E_Cd_/C_cr_, were all associated with both tubular proteinuria and reduced eGFR.

Tubular dysfunction and reduced eGFR were more prevalent in the diabetic than the non-diabetic groups by 8.7-fold (*p* < 0.001) and by 3-fold (*p* = 0.012), respectively. For a one-year increase in age, the POR for tubular dysfunction and reduced eGFR rose by 10% (*p* < 0.001) and 15% (*p* < 0.001), respectively. The doubling of E_Cd_/C_cr_ was associated with a 15% increase in the POR for tubular dysfunction (*p* = 0.002) and a 50% increase in the POR for reduced eGFR (*p* < 0.001).

### 3.3. Effects of Cadmium and Diabetes on β_2_M Excretion

We further evaluated the effects of diabetes and Cd exposure on E_β2M_ by scatterplots and covariance analyses (Figure 1).

A direct relationship was seen between E_β2M_/C_cr_ and E_Cd_/C_cr_ in the non-diabetic group (R^2^ 0.136, *p* < 0.001) (Figure 1a). After controlling for interactions and differences in age and BMI, the mean log[(E_β2M_/C_cr_) × 10^4^] was, respectively, the highest, in the middle, and lowest in non-diabetics who had high, moderate, and low E_Cd_/C_cr_ ranges (*F* = 24.08, *p* < 0.001) (Figure 1b).

In the analysis, including all 81 diabetics (Figure 1a), the relationship between E_β2M_/C_cr_ and E_Cd_/C_cr_ did not reach a statistically significant level (*p* = 0.053), and the variation in the mean log[(E_β2M_/C_cr_) × 10^4^] across the three E_Cd_/C_cr_ ranges was insignificant (*F* = 0.204, *p* = 0.816) (Figure 1b). However, the relationship between E_β2M_ and E_Cd_ was significant when 11 diabetic cases from a high-Cd exposure area were excluded (R^2^ 0.127, *p* = 0.002 (Figure 1c). After adjustment for age and BMI, the mean log[(E_β2M_/C_cr_) × 10^4^] was higher in diabetic subjects of low and moderate ranges of E_Cd_/C_cr_ than in non-diabetics of similar E_Cd_/C_cr_ ranges (Figure 1d). For the highest E_Cd_/C_cr_ subsets, the mean log[(E_β2M_/C_cr_) × 10^4^] in the diabetics and non-diabetics was similar.

The results of the regression analyses of β_2_M excretion are provided separately for the diabetics, diabetics from a low-exposure area, and non-diabetics (Table 3).

In a model including 70 diabetic cases from the low-exposure locality (Pakpoon), 24.4% of the variation in E_β2M_/C_cr_ was explained by all six independent variables. E_β2M_/C_cr_ varied directly with E_Cd_/C_cr_ (β = 0.375, *p* = 0.001) and obesity (β = 0.273, *p* = 0.015). However, when all 81 diabetics were included in the analysis, only 8.9% of the E_β2M_/C_cr_ variation was explained by age, log_2_[(E_Cd_/C_cr_) × 10^5^], smoking, obesity, gender, and hypertension. In effect, none of these six variables showed a significant association with E_β2M_/C_cr_ in the diabetic group.

In the non-diabetic group, the six independent variables explained 38.6% of the E_β2M_/C_cr_ variability, where E_β2M_/C_cr_ varied directly with age (β = 0.458, *p* < 0.001) and with E_Cd_/C_cr_ (β = 0.269, *p* < 0.001).

### 3.4. Inverse Association of eGFR and Cadmium

Similarly, we used multiple regression analyses to compare the strength of the association of the eGFR and E_Cd_/C_cr_ in three subsets (Table 4).

In a model including all diabetics, eGFR was inversely associated with age (β = −0.444) and E_Cd_/C_cr_ (β = −0.244), and these two variables, plus smoking, obesity, gender, and hypertension accounted for 33% of the eGFR variability. These six independent variables explained 29.3% of the total variation in eGFR among diabetics from a low-Cd-exposure area, and only age showed a significant association with eGFR reduction (β = −0.472). An association between the eGFR and E_Cd_/C_cr_ was insignificant (β = −0.145, *p* = 0.167).

In the non-diabetic group, age, E_Cd_/C_cr_, smoking, obesity, gender, and hypertension together accounted for 53.4% of the total eGFR variation. Distinct from the diabetics from a low-exposure area, the eGFR among those without diabetes was inversely associated with age (β = −0.574) and E_Cd_/C_cr_ (β = −0.263).

### 3.5. Inverse Association of β2M Excretion and eGFR

To assess the association of E_β2M_/C_cr_ with the eGFR, we employed scatterplots and covariance analyses, where differences in age and BMI were adjusted together with interactions (Figure 2).

In the diabetic group, there was a strong inverse relationship between E_β2M_/C_cr_ and the eGFR (R^2^ 0.412, *p* < 0.001) (Figure 2a), and the eGFR explained 33.4% of the E_β2M_/C_cr_ variation across the three eGFR ranges (Figure 2b). A large proportion of the E_β2M_/C_cr_ variation was explained by a single variable, eGFR.

Similarly, an inverse relationship was seen between E_β2M_/C_cr_ and E_Cd_/C_cr_ in the non-diabetic group (R^2^ 0.400, *p* < 0.001) (Figure 2c). The mean log[(E_β2M_/C_cr_) × 10^4^] was the highest, in the middle, and lowest in those with an eGFR ≤ 60, 61–90, and >90 mL/min/1.73m^2^, respectively (Figure 2d). The eGFR explained only 10.3% of the E_β2M_/C_cr_ variation across these ranges of the eGFR (*F* = 24.48, *p* < 0.001). This variation in E_β2M_/C_cr_ attributable to the eGFR was smaller compared to the diabetic group (Figure 2b).

### 3.6. β2M Excretion as a Function of GFR and Kidney Cadmium Burden

Because the GFR showed a strong influence on β_2_M excretion (Figure 2), we next compared E_β2M_/C_cr_ in subsets with a normal or reduced eGFR across three E_Cd_/C_cr_ ranges (<1, 1–4.99 and ≥5 ng/L filtrate). The results of these analyses are shown in Figure 3.

Among 81 diabetics, 63 had normal eGFR, while 18 (22.2%) had reduced eGFR (Figure 3a,b). A direct relationship between log[(E_β2M_/C_cr_) × 10^4^] and log[(E_Cd_/C_cr_) × 10^5^] was seen only in the reduced eGFR group (R^2^ 0.493, *p* < 0.001). In covariance analysis (Figure 2b), the mean log[(E_β2M_/Ccr) × 10^4^] tended to be higher in the low eGFR subsets of all three E_Cd_/C_cr_ ranges. The overall mean log[(E_β2M_/C_cr_) × 10^4^] was higher in the low eGFR than that of the high eGFR group (*p* < 0.001).

Among the 454 non-diabetics, 395 had a normal eGFR, while 59 (13%) had a reduced eGFR (Figure 3c,d). Log[(E_β2M_/C_cr_) × 10^4^] rose with log[(E_Cd_/C_cr_) × 10^5^] in both the low and normal eGFR groups. Like the diabetics, the mean log[(E_β2M_/Ccr) × 10^4^] tended to be higher in the low eGFR subsets of all three E_Cd_/C_cr_ ranges. The overall mean log[(E_β2M_/C_cr_) × 10^4^] was higher in the reduced eGFR than in the normal eGFR group (*p* < 0.001).

## 4. Discussion

In the present study, we compared the severity of Cd-induced nephropathy in diabetics and non-diabetics living in low- and high-Cd-exposure areas of Thailand. The prevalence of a reduced eGFR (below 60 mL/min/1.73 m^2^) in our cohort was 11.4% which is higher than that reported in studies from Spain (7%) [38] and Taiwan (6.3%) [39], but in line with the global prevalence of CKD, which varies between 8% and 16% [40]. Approximately, one in five cohort participants (27.6%) had tubular dysfunction, based on the conventional E_β2M_/E_cr_ ≥ 300 µg/g creatinine criterion (Table 1). We found that BMI, hypertension, smoking, and gender were independent variables that showed no significant associations with the risk for tubular proteinuria or a reduced eGFR, but age, diabetes, and measured long-term Cd exposure (body burden) did (Table 2).

For every one-year increase in age, the risks of tubular dysfunction and a reduced eGFR increased by 10% and 15%, respectively. The doubling of the Cd body burden increased the risk of tubular dysfunction by 15% while raising the risk of reduced eGFR by 50%. Thus, Cd had a particularly strong effect on the GFR in this population. This is consistent with the results of many other studies, which have linked an elevated risk of a reduced eGFR to environmental exposure to Cd, including studies from the U.S. [11,17,18,19], Thailand [20] Guatemala [21], Myanmar [22], Taiwan [23], and Korea [24,25].

The influence of both Cd body burden and the eGFR on E_β2M_/C_cr_ were evident when participants were stratified by the eGFR and ranges of E_Cd_/C_cr_, a measure of Cd body burden. In the diabetic group, 33.4% of the variation in E_β2M_/C_cr_ was associated with the eGFR (Figure 2b). This is a very large variation in the E_β2M_/C_cr_ that was due to a single variable, eGFR. In comparison, the eGFR explained only 10.3% of the E_β2M_/C_cr_ variation in the non-diabetic group (Figure 2b). After adjustment for age and BMI, the mean values of E_β2M_/C_cr_ in the diabetic and non-diabetic subsets were found to be higher in those with reduced eGFR compared to the normal eGFR subsets of similar Cd body burden (Figure 3a,b).

Evidence for increased susceptibility to Cd-induced tubulopathy among the diabetics comes from covariate analysis, where a relationship between E_β2M_/C_cr_ and E_Cd_/C_cr_ was seen only in those from a low-Cd-exposure region (Figure 1, Table 3). These findings are in line with the published reports showing the high susceptibility to the nephrotoxicity of Cd among people with diabetes, as discussed below.

The Cadmibel study found that diabetics were more susceptible than non-diabetics to Cd-induced nephrotoxicity [7]. A similar observation followed in studies conducted in Sweden [8,9], the Australian Torres Strait, [10], the U.S. [11], and Korea [12]. Experimental studies have shown that nephropathy due to diabetes and Cd are magnified when both the metal and the disease are present. The injection of Cd-MT complexes into obese diabetic mice and non-obese littermates resulted in increased urinary excretion of proteins and calcium in both groups [41]. However, in the diabetic mice, the dose of Cd-MT required to induce proteinuria and calciuria was one-fourth of that required in the controls. Cd-MT induced glycosuria in both groups. Chinese hamsters with hereditary diabetes are also highly susceptible to Cd-induced nephrotoxicity [42]. In recent histopathological studies, kidney tubular degeneration and fibrosis due to Cd were more pronounced in diabetic than non-diabetic rats [43,44].

Increases in the risks of prediabetes and diabetes among U.S. adults have been associated with E_Cd_/E_cr_ of 1–2 µg/g creatinine [1,2]. In a community-based study in Dallas, Texas, an elevated risk of diabetes was linked to environmental Cd exposure [3]. In a meta-analysis of pooled data from 42 studies, the risks of prediabetes and diabetes increased linearly with blood and urinary Cd; the risk of prediabetes reached a plateau at an E_Cd_/E_cr_ rate of 2 µg/g creatinine, and the diabetic risk rose as blood Cd reached 1 µg/L [5,6]. Of note, these urinary Cd and blood Cd levels have also been in the range associated with a reduced eGFR in studies conducted in many countries listed previously. In a Chinese population study, dietary Cd exposure estimates of 23.2, 29.6, and 36.9 μg/d were associated with 1.73-, 2.93-, and 4.05-fold increments in the prevalence of CKD, compared to a 16.7 μg/d intake level [45]. A diet high in rice, pork, and vegetables was associated with a 4.56-fold increase in the prevalence of CKD [45].

Smoking has been shown to promote both the onset and progression of CKD [46,47]. In a meta-analysis of data from 104 studies, an increase in the odds of CKD of 18% was seen among current and former smokers compared to those who never smoked [46]. A Singaporean prospective cohort (*n* = 63,257, 30.6% were smokers) has implicated smoking in the progression of CKD [47]. With adjustment for confounders, smoking increased the risk of end-stage kidney disease by 29% compared to non-smokers, while the risk of kidney failure diminished after quitting smoking for more than 10 years [47]. A strong dose-dependent association was seen between the number of years of smoking and kidney failure [47].

In the present study, the risk of CKD (reduced eGFR) was not associated with smoking; instead, it was associated with an indicator of cumulative Cd exposure from all sources. Per the doubling of Cd body burden, there was a 50% increase in the prevalence odds of CKD (Table 2). Smoking is a significant source of Cd exposure, given that cigarette smoke contains Cd in volatile metallic and oxide (CdO) forms, which have transmission rates 5 to 10 times higher than those that enter through the gut [48]. Cd exposure through smoking has been found to increase the risk of diabetic nephropathy in a Dutch cross-sectional study, including 231 patients with type 2 diabetes, where active smokers were found to have significantly higher blood Cd compared to never smokers and former smokers [49]. Data also demonstrated that smoke-derived Cd mediated this nephrotoxicity [49]. In a six-year median follow-up of these 231 diabetic patients, both Cd and active smoking were associated with progressive eGFR reduction [50]. Collectively, findings from the Dutch cohorts support the premise that exposure to even low levels of environmental Cd promotes the development and progression of diabetic kidney disease. This lends support to our observation that people with diabetes are more susceptible to Cd-induced tubulopathy than non-diabetics.

## 5. Conclusions

This study shows that tubular dysfunction and a reduced eGFR are more severe and more prevalent in diabetics than non-diabetics of similar age, BMI, and Cd body burden. Public health resources that promote cessation of smoking and educate consumers about foods known to contain high levels of Cd are likely to have significant health benefits.

## Figures and Tables

**Figure 1 toxics-11-00390-f001:**
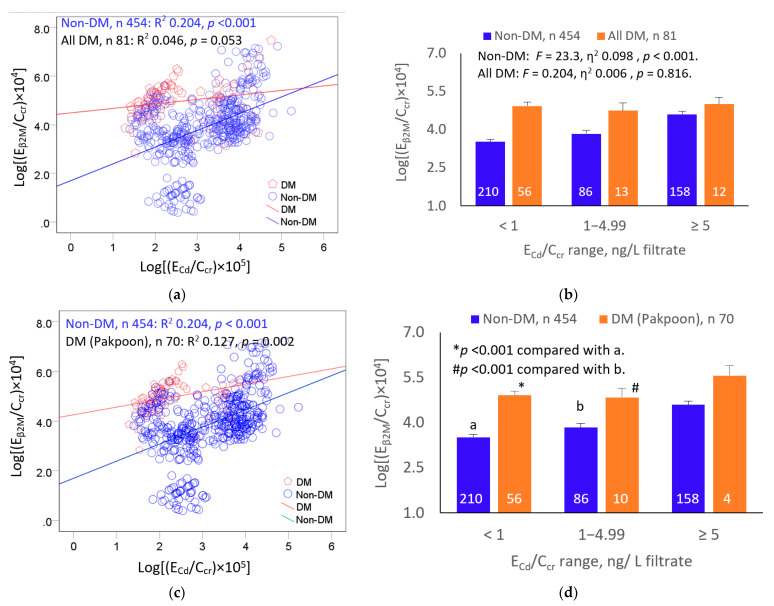
Effects of diabetes and cadmium exposure on β_2_M excretion. Scatterplot (**a**) relates log[(E_β2M_/C_cr_) × 10^4^] to log[(E_Cd_/C_cr_) × 10^5^] in all diabetic and all non-diabetic participants. Bar graph (**b**) depicts mean log[(E_β2M_/C_cr_) × 10^4^] in all diabetic and non-diabetics grouped by ranges of log[(E_Cd_/C_cr_) × 10^5^]. Scatterplot (**c**) relates log[(E_β2M_/C_cr_) × 10^4^] to log[(E_Cd_/C_cr_) × 10^5^] in diabetics from Pakpoon and all non-diabetics. Bar graph (**d**) depicts mean log[(E_β2M_/C_cr_) × 10^4^] in diabetics from Pakpoon and all non-diabetics grouped by ranges of log[(E_Cd_/C_cr_) × 10^5^]. Coefficients of determination (R^2^) and *p*-values are provided for all scatterplots. Mean values were adjusted for age, BMI, and interactions. Units of E_β2M_/C_cr_ and E_Cd_/C_cr_ are ng/L of filtrate.

**Figure 2 toxics-11-00390-f002:**
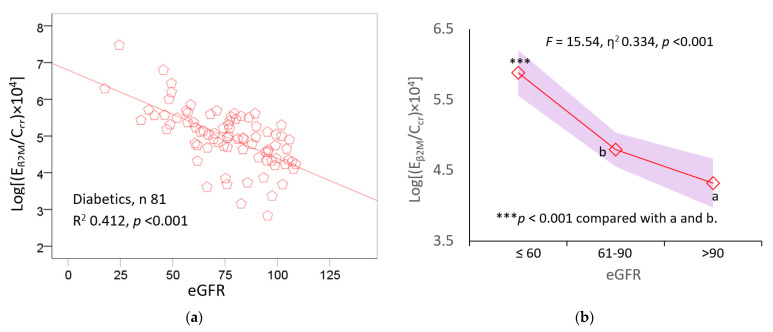
Excretion rates of β_2_M in relation to eGFR reduction. Scatterplot (**a**) relates log[(E_β2M_/C_cr_) × 10^4^] to eGFR in all diabetics. Graph (**b**) depicts mean log[(E_β2M_/C_cr_) × 10^4^] and the variability of each mean in diabetics grouped by ranges of eGFR. Scatterplot (**c**) relates log[(E_β2M_/C_cr_) × 10^4^] to eGFR among non-diabetics. Graph (**d**) depicts mean log[(E_β2M_/C_cr_) × 10^4^] and the variability of each mean in diabetics grouped by ranges of eGFR. Coefficients of determination (R^2^) and *p*-values are provided for all scatterplots. Mean values were adjusted for covariates and interactions. Unit of E_β2M_/C_cr_ is µg/L of filtrate, and the unit of eGFR is mL/min/1.73 m^2^.

**Figure 3 toxics-11-00390-f003:**
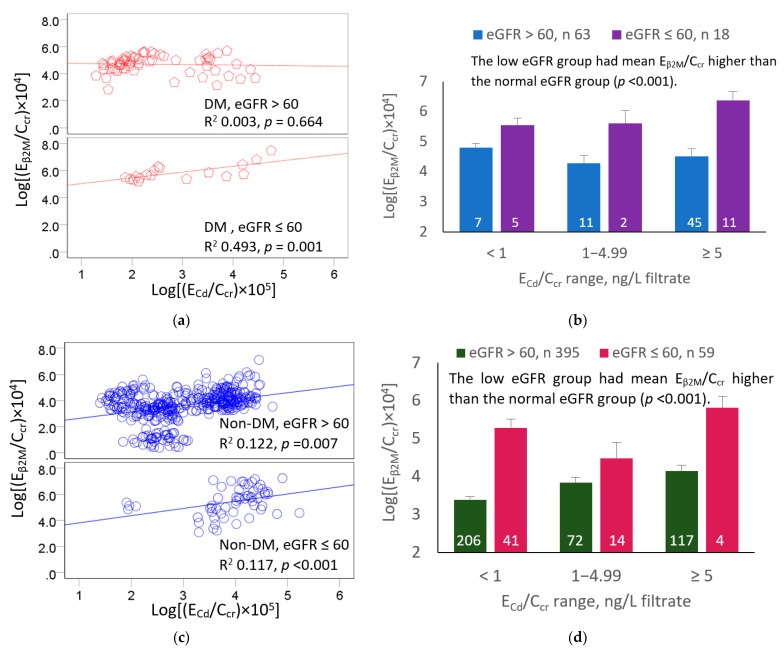
Increments of β_2_M excretion as a function of kidney cadmium burden and GFR. Scatterplot (**a**) relates log[(E_β2M_/C_cr_) × 10^4^] to log[(E_Cd_/C_cr_) × 10^5^] in diabetics grouped by eGFR levels > 60 and ≤60 mL/min/1.73 m^2^. Bar graph (**b**) depicts mean log[(E_β2M_/C_cr_) × 10^4^] in diabetics grouped by eGFR and ranges of E_Cd_/C_cr_. Scatterplot (**c**) relates log[(E_β2M_/C_cr_) × 10^4^] to log[(E_Cd_/C_cr_) × 10^5^] in non-diabetics grouped by eGFR levels > 60 and ≤60 mL/min/1.73 m^2^. Graph (**d**) depicts mean log[(E_β2M_/C_cr_) × 10^4^] in non-diabetics grouped by eGFR and ranges of E_Cd_/C_cr_. Coefficients of determination (R^2^) and *p*-values are provided for all scatterplots. Mean values were adjusted for age and BMI differences, and interactions. Units of E_β2M_/C_cr_ and E_Cd_/C_cr_ are ng/L of filtrate, and the unit of eGFR is mL/min/1.73 m^2^.

**Table 1 toxics-11-00390-t001:** Characterization of cohort participants.

Parameters	All Participants, *n* = 674	Diabetics, *n* = 81	Non-Diabetics, *n* = 593	*p*
Low-exposure locality (%)	67.1	86.4	64.4	<0.001
Female, %	57.6	74.1	55.3	<0.001
Smoking, %	31.9	14.8	34.2	<0.001
Hypertension ^c^ %	25.8	55.6	21.8	<0.001
Age, years	47.1 ± 16.3	58.5 ± 9.7	45.6 ± 16.4	<0.001
BMI, kg/m^2^	22.8 ± 4.2	26.0 ± 4.8	22.3 ± 3.9	<0.001
Obese ^a^ (%)	3.9		2.4	<0.001
eGFR ^b^, mL/min/1.73 m^2^	87.4 ± 23.1	76.2 ± 21.0	89.0 ± 23.0	<0.001
Reduced eGFR ^c^ (%)	11.4	22.2	9.9	<0.001
Plasma creatinine, mg/dL	0.94 ± 0.28	0.96 ± 0.36	0.93 ± 0.27	0.911
Urine creatinine, mg/dL	98.8 ± 68.4	90.1 ± 58.2	99.9 ± 69.7	0.453
Plasma-to-urine creatinine ratio	0.016 ± 0.015	0.016 ± 0.012	0.016 ± 0.015	0.391
Urine Cd, µg/L	4.59 ± 11.92	2.75 ± 6.94	4.84 ± 12.44	0.003
Urine β_2_M, µg/L	2327 ± 11,517	4105 ± 21,667	2010 ± 8532	<0.001
Normalized to C_cr_ (E_x_/C_cr_) ^d^				
(E_Cd_/C_cr_) × 100, µg/L filtrate	4.48 ± 10.38	3.32 ± 8.40	4.64 ± 10.62	<0.001
(E_β2M_/C_cr_) × 100, µg/L filtrate	3858 ± 20,142	6694 ± 33,617	3352 ± 16,642	<0.001
(E_β2M_/C_cr_) × 100, µg/L filtrate (%)				
≥300	36.3	75.3	29.3	<0.001
≥1000	20.4	49.4	14.8	<0.001
Normalized to E_cr_ (E_x_/E_cr_) ^e^				
E_Cd_/E_cr_, µg/g creatinine	4.12 ± 7.80	2.75 ± 5.52	4.30 ± 8.04	<0.001
E_β2M_/E_cr_, µg/g creatinine	2540 ± 11,420	4004 ± 13,978	2279 ± 10,899	<0.001
E_β2M_/E_cr_, µg/g creatinine (%)				
≥300	37.6	79.0	30.2	<0.001
≥1000	21.3	53.1	15.6	<0.001

Abbreviations: *n*, number of subjects; BMI, body mass index; β_2_M, β_2_-microglobulin; eGFR, estimated glomerular filtration rate; E_x_, excretion of x; cr, creatinine; C_cr_, creatinine clearance; Cd, cadmium; ^a^ Obese was defined as BMI > 30 kg/m^2^; ^b^ eGFR, was determined by CKD-EPI equations [28]; ^c^ reduced eGFR corresponds to eGFR ≤ 60 mL/min/1.73m^2^; ^d^ E_x_/E_cr_ = [x]_u_/[cr]_u_; ^e^ E_x_/C_cr_ = [x]_u_[cr]_p_/[cr]_u_, where x = β_2_M or Cd [29]. Data for all continuous variables are arithmetic means ± standard deviation (SD). Data for β_2_M were from 535 subjects. Data for all other continuous variables were from 674 subjects. For all tests, *p* ≤ 0.05 identifies statistical significance, determined by Pearson chi-square test for % differences and by the Mann–Whitney U-test for mean differences between diabetes and non-diabetes.

**Table 2 toxics-11-00390-t002:** Risk factors for tubular dysfunction and reduced eGFR.

Parameters	Number of Participants	Tubular Dysfunction ^a^		Reduced eGFR ^b^	
		POR (95% CI)	*p*	POR (95% CI)	*p*
Age, year	535 (674)	1.100 (1.076, 1.126)	<0.001	1.146 (1.107, 1.188)	<0.001
BMI, kg/m^2^	535 (674)	0.963 (0.908, 1.021)	0.202	1.070 (0.993, 1.153)	0.075
Log_2_[(E_Cd_/C_cr_) × 10^5^]	535 (674)	1.149 (1.054, 1.253)	0.002	1.500 (1.304, 1.724)	<0.001
Hypertension	162 (174)	0.724 (0.444, 1.181)	0.195	0.744 (0.387, 1.428)	0.374
Smoking	195 (215)	1.563 (0.863, 2.830)	0.140	1.124 (0.511, 2.474)	0.771
Gender (male)	223 (286)	0.819 (0.480, 1.398)	0.464	1.284 (0.615, 2.680)	0.506
Diabetes	81 (81)	8.690 (4.421, 17.08)	<0.001	2.973 (1.274, 6.937)	0.012

Abbreviations: POR, prevalence odds ratio; CI, confidence interval. Coding: female = 1, male = 2, normotension = 1, hypertension = 2, non-smoker = 1, smoker = 2. ^a^ Tubular dysfunction was defined as (E_β2M_/C_cr_) × 100 ≥ 300 µg/L filtrate; ^b^ reduced eGFR was defined as estimated GFR ≤ 60 mL/min/1.73 m^2^. Data were generated from logistic regression analyses relating POR to tubular proteinuria and reduced eGFR to a set of seven independent variables (first column). First number and number in parenthesis in second column correspond to number of participants in analyses of tubular dysfunction and reduced eGFR, respectively. For all tests, *p*-values ≤ 0.05 indicate a statistically significant association of POR with a given independent variable.

**Table 3 toxics-11-00390-t003:** Association of β_2_M excretion with cadmium excretion and other independent variables.

Independent Variables/Factors		Excretion Rate of β_2_M ^a^				
	All Diabetics, *n* = 81		Diabetics, *n* = 70		Non-Diabetics, *n* = 454	
	β ^b^	*p*	β	*p*	β	*p*
Age, years	0.202	0.072	0.159	0.159	0.458	<0.001
Log_2_[(E_Cd_/C_cr_) × 10^5^], µg/L filtrate	0.181	0.113	0.375	0.001	0.269	<0.001
Smoking	0.206	0.155	0.253	0.055	0.036	0.443
Obesity	0.145	0.194	0.273	0.015	−0.049	0.193
Gender	−0.136	0.328	−0.209	0.122	0.025	0.572
Hypertension	0.105	0.346	−0.032	0.767	0.024	0.532
Adjusted R^2^	0.089	0.043	0.244	0.001	0.386	<0.001

Abbreviations: *n*, number of participants. ^a^ Excretion rate of β_2_M as log[(E_β2M_/C_cr_) × 10^4^]; ^b^ β, standardized regression coefficients. Coding: female = 1, male = 2, normotension = 1, hypertension = 2, non-smoker = 1, smoker = 2, obese = 1, non-obese = 2. Data were generated from multiple regression model analyses relating E_β2M_/C_cr_ to six independent variables (first column) in all participants, diabetes, and non-diabetes. For all tests, *p*-values < 0.05 indicate a statistically significant association. β coefficients indicate the strength of the association of E_β2M_/C_cr_ and independent variables. Adjusted R^2^ indicates the proportion of the variation in E_β2M_/C_cr_ attributable to all six independent variables.

**Table 4 toxics-11-00390-t004:** Associations of eGFR with cadmium excretion and other independent variables.

Independent Variables/Factors		eGFR ^a^, mL/min/1.73m^2^				
	All Diabetics, *n* = 81		Diabetics, *n* = 70		Non-Diabetics, *n* = 593	
	β ^b^	*p*	β	*p*	β	*p*
Age, year	−0.444	<0.001	−0.472	<0.001	−0.574	<0.001
Log_2_[(E_Cd_/C_cr_) × 10^5^], µg/L filtrate	−0.244	0.014	−0.145	0.167	−0.263	<0.001
Smoking	−0.208	0.095	−0.175	0.167	0.033	0.352
Obesity	−0.151	0.116	−0.146	0.170	0.037	0.190
Gender	0.163	0.173	0.162	0.214	−0.024	0.448
Hypertension	−0.045	0.634	−0.066	0.530	−0.014	0.633
Adjusted R^2^	0.330	<0.001	0.293	<0.001	0.534	<0.001

Abbreviations: *n*, number of participants. ^a^ eGFR was determined by CKD-EPI equations [28]; ^b^ β, standardized regression coefficients. Coding: female = 1, male = 2, normotension = 1, hypertension = 2, non-smoker = 1, smoker = 2. Data were generated from multiple regression model analyses relating eGFR to six independent variables (first column) in all participants, diabetes, and non-diabetes. For all tests, *p*-values < 0.05 indicate a statistically significant association. β coefficients indicate the strength of the association of the eGFR and independent variables. Adjusted R^2^ indicates the proportion of the variation in eGFR attributable to all six independent variables.

## Data Availability

All data are contained within this article.

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
