# Peer review of "Cadmium-Induced Tubular Dysfunction in Type 2 Diabetes: A Population-Based Cross-Sectional Study"

_toxics, 2023, doi:10.3390/toxics11040390_

Round 1

Reviewer 1 Report

This manuscript is an interesting article that reveals a correlation between diabetes and tubular proteinuria and reduced eGFR in residents of a Cd-low pollution area in Thailand. However, there are some parts that need to be added or corrected, please see below.

1.      Cohort studies have been conducted on residents in Cd-low and Cd-high exposure areas, but please provide the specific levels of Cd contamination in each area. Are Cd concentrations in soil or Cd concentrations in crops produced in each region used as a standard for exposure indices?

2.      In the Conclusion, the authors mention that smoking cessation is encouraged. In general, I believe that smoking cessation is encouraged. However, the results of the current study show no significant association between smoking and diabetes, tubular dysfunction, and decreased eGFR. I think the authors are assuming exposure to Cd in cigarettes, so I think you need to explain this a little more.

Author Response

Comments and Suggestions - REVIEWER 1

General comment. This manuscript is an interesting article that reveals a correlation between diabetes and tubular proteinuria and reduced eGFR in residents of a Cd-low pollution area in Thailand. However, there are some parts that need to be added or corrected, please see below.

General response: We thank this reviewer for evaluating our manuscript and helpful suggestions. Our manuscript has been revised accordingly. We provided below point-by-point response to each comment.  Changes to the text are in blue.

Comment 1. Cohort studies have been conducted on residents in Cd-low and Cd-high exposure areas, but please provide the specific levels of Cd contamination in each area. Are Cd concentrations in soil or Cd concentrations in crops produced in each region used as a standard for exposure indices?

Response 1. We have provided further information regarding the environmental sources of exposure to Cd in Section 2.1 (lines 89-97).

“As workplace exposure was an exclusion criterion, participants presumably acquired Cd from the diet and/or smoking. Based on the measured levels of Cd in duplicate diets [32] and a nation-wide survey of Cd levels in soils and food crops [33], environmental exposure to Cd in Bangkok and Pakpoon were low, and the Mae Sot district was high. From a previous study conducted in the Mae Sot district, the Cd content of the paddy soil samples exceeded the standard of 0.15 mg/kg, and rice samples collected from households contained 4-time the permissible Cd level of 0.1 mg/kg [34]. A health survey reported the prevalence of CKD in the Mae Sot was 16.1%, while the prevalence of tubular proteinuria was 36.1% [35].”

[32] Zhang, Z.W.; Shimbo, S.; Watanabe, T.; Srianujata, S.; Banjong, O.; Chitchumroonchokchai, C.; Nakatsuka, H.; Matsuda-Inoguchi, N.; Higashikawa, K.; Ikeda, M. Non-occupational lead and cadmium exposure of adult women in Bangkok, Thailand. Sci. Total. Environ. 1999, 226, 65-74.

[33] Zarcinas, B.A.; Pongsakul, P.; McLaughlin, M.J.; Cozens, G. Heavy metals in soils and crops in Southeast Asia. 2. Thailand. Environ. Geochem. Health 2004, 26, 359-371.

[34] Suwatvitayakorn, P.; Ko, M.S.; Kim, K.W.; Chanpiwat, P. Human health risk assessment of cadmium exposure through rice consumption in cadmium-contaminated areas of the Mae Tao sub-district, Tak, Thailand. Environ. Geochem. Health 2020, 42, 2331-2344.

[35] Swaddiwudhipong, W.; Nguntra, P.; Kaewnate, Y.; Mahasakpan, P.; Limpatanachote, P.; Aunjai, T.; Jeekeeree, W.; Punta, B.; Funkhiew, T.; Phopueng, I. Human health effects from cadmium exposure: comparison between persons living in cadmium-contaminated and non-contaminated areas in northwestern Thailand. Southeast Asian J. Trop. Med. Publ. Health, 2015, 46, 133-142.

Comment 2. In the Conclusion, the authors mention that smoking cessation is encouraged. In general, I believe that smoking cessation is encouraged. However, the results of the current study show no significant association between smoking and diabetes, tubular dysfunction, and decreased eGFR. I think the authors are assuming exposure to Cd in cigarettes, so I think you need to explain this a little more.

Response 2: We have now included new paragraphs and new references in the Discussion (lines 377-400) quoted below to support our assertion to discourage smoking which is also a source of Cd exposure. In addition, evidence that Cd mediated an effect of smoking on eGFR is provided.

“Smoking has been shown to promote both the onset and progression of CKD [46,47]. In a meta-analysis of data from 104 studies, an increase in odds of CKD of 18% was seen among current and former smokers, compared to those who never smoked [46]. A Singaporean prospective cohort (n = 63,257, 30.6% were smokers) has implicated smoking in the progression of CKD [47]. With adjustment for confounders, smoking increased the risk of end-stage kidney disease by 29%, compared to non-smokers, while the risk of kidney failure fell after quitting smoking for more than 10 years [47]. A strong dose-dependent association was seen between number of years of smoking and kidney failure [47].

In the present study, the risk of CKD (reduced eGFR) was not associated smoking, but instead it was associated with an indicator of cumulative Cd exposure from all sources (POR = 1.610, p <0.001) (Table 2). Smoking is a significant source of Cd exposure, given that cigarette smoke contains Cd in volatile metallic and oxide (CdO) forms which have transmission rates 5- to 10-times higher than those enter through the gut [48]. Cd exposure through smoking has been found to increase the risk of diabetic nephropathy. In a Dutch cross-sectional study, which included 231 patients with type 2 diabetes, active smokers had significantly higher blood Cd, compared to never smokers, and former smokers [49]. Data also demonstrated that smoke derived Cd mediated this nephrotoxicity [49]. In a six-year median follow-up of these 231 diabetic patients, both Cd and active smoking were associated with progressive eGFR reduction [50]. Collectively, findings from the Dutch cohorts support the premise that exposure to even low levels of environmental Cd promotes the development and progression of diabetic kidney disease. This lends support to our observation that people with diabetes are more susceptible to Cd-induced tubulopathy than the non-diabetics.”

[46] Kelly, J.T.; Su, G.; Zhang, L.; Qin, X.; Marshall, S.; González-Ortiz, A.; Clase, C.M.; Campbell, K.L.; Xu, H.; Carrero, J.J. Modifiable lifestyle factors for primary prevention of CKD: A systematic review and meta-analysis. J. Am. Soc. Nephrol. 2021, 32, 239-253.

[47] Jin, A.; Koh, W.P.; Chow, K.Y.; Yuan, J.M.; Jafar, T.H. Smoking and risk of kidney failure in the Singapore Chinese health study. PLoS One 2013, 8, e62962.

[48] Pappas, R.S.; Fresquez, M.R.; Watson, C.H. Cigarette smoke cadmium breakthrough from traditional filters: Implications for exposure. J. Anal. Toxicol. 2015, 39, 45-51.

[49] Hagedoorn, I.J.M.; Gant, C.M.; Huizen, S.V.; Maatman, R.G.H.J.; Navis, G.; Bakker, S.J.L.; Laverman, G.D. Lifestyle-related exposure to cadmium and lead is associated with diabetic kidney disease. J. Clin. Med. 2020, 9, 2432.

[50] Oosterwijk, M.M.; Hagedoorn, I.J.M.; Maatman, R.G.H.J.; Bakker, S.J.L.; Navis, G.; Laverman, G.D. Cadmium, active smoking and renal function deterioration in patients with type 2 diabetes. Nephrol. Dial. Transplant 2022, gfac270.

Reviewer 2 Report

The authors investigated the influence on Cd exposure in type 2 diabetes-induced nephropathy in Thailand as an epidemiological study. In my opinion, although such an epidemiological study is important for understanding the relationship between the onset and progression of diseases and the exposure to hazardous chemicals, the present set of data are not conclusive enough to draw unambiguous structural conclusions. The overall structure of the paper should be reconsidered. I therefore recommend a reject of the paper at this time.

1. Although this paper gives the impression the results obtained in the study are representative for the whole of Thailand, I think this is an overstatement. It should be stated as if it is only a comparison between polluted and low-polluted areas in Thailand, and even so, the significance of this study would not be diminished.

2. Regarding the above, the sources of Cd in the polluted area should be described. In the Conclusion section, the authors described the importance of the restrictions of smoking and diet, but is there excessive smoking or food contamination present in the area? On the other hand, the possibility of exposure due to environmental contamination, such as water, should be considered.

3. One of the primary complications of diabetes is nephropathy, and cadmium causes nephrotoxicity, especially proximal tubular damage. The author should describe whether the severity of DM-induced nephropathy was caused by cadmium in an additive or synergistic manner. In Table 3 and 4, the value of eGFR and B2M in non-DM correlate with Cd, but not that in DM, suggesting that the mechanisms of Cd and DM-induced nephrotoxicity are independent.

4. The bioaccumulation of Cd increases with age. All models should be adjusted for age, and other correlated factors such as sex.

5. line 63: “metallothionine (MT)” should be written as “metallothionein (MT).”

6. Figure 1a and 1c: the straight lines are so thin that it is difficult to tell the difference in color.

7. Figure 2b and 2d: since the numbers on the horizontal axis do not correspond to 2a and 2c, which confuses the readers. Generally, the values on the horizontal axis should be increased from left to right. Additionally, the horizontal axis in Figure 2b is misrepresented and should be corrected.

Author Response

Comments and Suggestions – REVIEWER 2

General comment. The authors investigated the influence on Cd exposure in type 2 diabetes-induced nephropathy in Thailand as an epidemiological study. In my opinion, although such an epidemiological study is important for understanding the relationship between the onset and progression of diseases and the exposure to hazardous chemicals, the present set of data are not conclusive enough to draw unambiguous structural conclusions. The overall structure of the paper should be reconsidered. I therefore recommend a reject of the paper at this time.

General response: We thank this reviewer for evaluating our manuscript and helpful suggestions. In accordance with the Reviewer guidance, our manuscript has undergone extensive revisions. We provided below point-by-point response to each comment.  Changes to the text are in blue.

Comment 1. Although this paper gives the impression the results obtained in the study are representative for the whole of Thailand, I think this is an overstatement. It should be stated as if it is only a comparison between polluted and low-polluted areas in Thailand, and even so, the significance of this study would not be diminished.

Response 1: The title of our paper has now been changed to “Tubular Dysfunction Is Significantly Worse in Type 2 Diabetics Exposed to Low Levels of Environmental Cadmium”. 

Study objective statements have been modified, and part of the abstract has been rewritten to reflect our basic study design and new conclusion.

Objective

Therefore, the present study aimed to quantify Cd exposure levels, kidney tubular dysfunction and reduction of eGFR, experienced by diabetics and non-diabetics who lived in low- and high-Cd exposure areas of Thailand.

Abstract

“However, little is known about the nephrotoxicity of Cd in the diabetic population. Here, we compared Cd exposure, eGFR and tubular dysfunction in both diabetics (n = 81) and non-diabetics (n = 903) who were residents in low- and high-Cd exposure areas of Thailand”.

“A reduction in GFR and tubular dysfunction were more severe in diabetics than non-diabetics of similar age, BMI, and ranges of Cd body burden.  It thus appears that diabetics are more susceptible to Cd-induced nephropathy than non-diabetics.

Comment 2. The sources of Cd in the polluted area should be described. In the Conclusion section, the authors described the importance of the restrictions of smoking and diet, but is there excessive smoking or food contamination present in the area? On the other hand, the possibility of exposure due to environmental contamination, such as water, should be considered.

Response 2: We have provided further information regarding the environmental sources of Cd exposure among cohort participants in Section 2.1 (lines 89-97).

“As workplace exposure was an exclusion criterion, participants presumably acquired Cd from the diet and/or smoking. Based on the measured levels of Cd in duplicate diets [32] and a nation-wide survey of Cd levels in soils and food crops [33], environmental exposure to Cd in Bangkok and Pakpoon were low, and the Mae Sot district was high. From a previous study conducted in the Mae Sot district, the Cd content of the paddy soil samples exceeded the standard of 0.15 mg/kg, and rice samples collected from households contained 4-time the permissible Cd level of 0.1 mg/kg [34]. A health survey reported the prevalence of CKD in the Mae Sot was 16.1%, while the prevalence of tubular proteinuria was 36.1% [35].”

[32] Zhang, Z.W.; Shimbo, S.; Watanabe, T.; Srianujata, S.; Banjong, O.; Chitchumroonchokchai, C.; Nakatsuka, H.; Matsuda-Inoguchi, N.; Higashikawa, K.; Ikeda, M. Non-occupational lead and cadmium exposure of adult women in Bangkok, Thailand. Sci. Total. Environ. 1999, 226, 65-74.

[33] Zarcinas, B.A.; Pongsakul, P.; McLaughlin, M.J.; Cozens, G. Heavy metals in soils and crops in Southeast Asia. 2. Thailand. Environ. Geochem. Health 2004, 26, 359-371.

[34] Suwatvitayakorn, P.; Ko, M.S.; Kim, K.W.; Chanpiwat, P. Human health risk assessment of cadmium exposure through rice consumption in cadmium-contaminated areas of the Mae Tao sub-district, Tak, Thailand. Environ. Geochem. Health 2020, 42, 2331-2344.

[35] Swaddiwudhipong, W.; Nguntra, P.; Kaewnate, Y.; Mahasakpan, P.; Limpatanachote, P.; Aunjai, T.; Jeekeeree, W.; Punta, B.; Funkhiew, T.; Phopueng, I. Human health effects from cadmium exposure: comparison between persons living in cadmium-contaminated and non-contaminated areas in northwestern Thailand. Southeast Asian J. Trop. Med. Publ. Health, 2015, 46, 133-142.

Comment 3. One of the primary complications of diabetes is nephropathy, and cadmium causes nephrotoxicity, especially proximal tubular damage. The author should describe whether the severity of DM-induced nephropathy was caused by cadmium in an additive or synergistic manner. In Table 3 and 4, the value of eGFR and B2M in non-DM correlate with Cd, but not that in DM, suggesting that the mechanisms of Cd and DM-induced nephrotoxicity are independent.

Response 3: To provide evidence that Cd influenced β2M excretion and/or GFR differently in diabetics and non-diabetics, the results of two additional multiple regression models have been added to Table 3 and 4. These additional analyses indicate that the effect of Cd on β2M was significant only in low body burden (ECd/Ccr <5 ng/L filtrate). As our study is a cross-sectional design, it is not possible to separate the effect of DM from that of Cd. We cautiously interpret these results to suggest that diabetics are more susceptible to the nephrotoxicity of Cd. Two recent histopathological studies using rats have been inserted (lines 361-363) to support this interpretation.

We believe that a prospective case-control study design is required to address the mechanistic aspects. We discuss data from the Netherlands that unambiguously show the influence of smoking and Cd exposure on the onset and progression of diabetic kidney disease using case-control study design (lines 386-392). Notably, however, only eGFR was only kidney parameter considered in the Netherlands studies [49,50]. In comparison, eGFR and tubulopathy both were investigated in our study.

49.Hagedoorn, I.J.M.; Gant, C.M.; Huizen, S.V.; Maatman, R.G.H.J.; Navis, G.; Bakker, S.J.L.; Laverman, G.D. Lifestyle-related exposure to cadmium and lead is associated with diabetic kidney disease. J. Clin. Med. 2020, 9, 2432.

  1. Oosterwijk, M.M.; Hagedoorn, I.J.M.; Maatman, R.G.H.J.; Bakker, S.J.L.; Navis, G.; Laverman, G.D. Cadmium, active smoking and renal function deterioration in patients with type 2 diabetes. Nephrol. Dial. Transplant 2022, gfac270.

Comment 4. The bioaccumulation of Cd increases with age. All models should be adjusted for age, and other correlated factors such as sex.

Response 4: The mean β2M excretion rate and the mean eGFR presented in the original manuscript were all adjusted for age and BMI. Due to the small sample size of diabetic cases, parameters other than age and BMI have not been adjusted for.    

Comment 5. line 63: “metallothionine (MT)” should be written as “metallothionein (MT).”

Response 5. A typo mistake has been corrected.

Comment 6. Figure 1a and 1c: the straight lines are so thin that it is difficult to tell the difference in color.

Response 6. The thickness of regression lines in Figures 1a and 1c have been increased so that they are now clearer.

Comment 7. Figure 2b and 2d: since the numbers on the horizontal axis do not correspond to 2a and 2c, which confuses the readers. Generally, the values on the horizontal axis should be increased from left to right. Additionally, the horizontal axis in Figure 2b is misrepresented and should be corrected.

Response 7. Changes to Figures 2b and 2d have been made as advised, and the error in labeling of the horizonal axis has been rectified. 

Reviewer 3 Report

In general, there are five major remarks. First one, the methodology: three different methods for determining the Cd level used on different groups of subjects, significant disproportion in the number of subjects exposed to higher concentrations of Cd between diabetic and non-diabetic group, lack of the exact measurement of the plasma Cd concentrations, huge disproportion between the total number of non-diabetics and diabetics, differences between the groups in their demographic characteristics which have appeared to significantly affect multiple correlation analysis aiming to measure the strength of the dependence of risk from proteinuria and estimated glomerular filtration with the excreted concentrations of Cd. Due to these flaws the conclusion is overemphasized and compromised by the effect of those flaws on the overall results.

To retain the manuscript being further considered for publication it is mandatory that the authors recheck their exclusion and inclusion criteria, homogenize two groups of subjects, make them comparable in all parameters, even the extent of exposure to Cd and number of participants. After that has been done, they should rerun their statistical analyses and take into a consideration uncertainty factor occurring from the use of different methods in Cd concentration determination in different subjects.

Abstract

Line 28 – it is not clear how did the authors reach that conclusion. There is no mentioning of correlation analysis between Cd plasma content and clearance or proteinuria. The authors provide results of CDK biomarkers between subjects with and without diabetes, but, as they said, of the similar Cd clearance in both groups. How did the authors differentiate diabetes induced CDK and Cd induced CDK?

Materials and methods

Relevance of the Cd excretion level as the measure for Cd burdening of the organism is questionable in subjects with impaired or even diminished kidney function, which is the case in the diabetes subjects. As the authors mentioned, metallothionein chelated Cd is a measure of kidney damage. Thus, diabetes patients excreting chelated Cd and those excreting non-chelated one cannot be grouped together, and their results considered jointly in the comparison to the results of non-diabetic subjects. Further, it can not be deduced whether the CDK in those chelated Cd excreting patients is affected by diabetes or Cd, because there was not any comparison in the serum Cd levels between CDK-developed diabetics and diabetics with retained normal kidney function.

For three groups of subjects three different methods have been used to determine urine Cd concentration. The authors stated that each method was calibrated by the standard sample, but they do not mention whether or what is they ensured that Cd values measured by ICP-MS in a urine sample would give the same results if the same urine sample would be subject to AAS detection of Cd.

Diabetics and non-diabetics groups significantly differ in the number of subjects being exposed to low levels of Cd, being in favor of diabetics. This fact additionally undermines the conclusion provided by the authors regarding the Cd adverse effect on kidneys in diabetics. Besides, there is significant difference in the age, the group consisting of diabetic patients is older compared to the reference group. Furthermore, in the Table 2, Cd level is not identified as a risk factor affecting the prevalence odd ratio for proteinuria or estimated glomerular filtration.

Regarding the results presented in Figure 1, particularly 1b and 1d, it is difficult to draw any conclusions by comparing results for 11 or 4 diabetics with those for 237 or327 non-diabetics, respectively. Again, regardless of the adjusted Cd excretion, in the group of diabetic suffering subjects no difference in β2M excretion as a biomarker for proteinuria was recorded.

Multiple regression analysis showed significant correlation between excreted Cd and proteinuria risk in the non-diabetic group. However, the result is significantly affected by other group characteristics, and which most interesting invert effect of the blood pressure on β2M excretion was observed, which is unusual.

Significant correlation between the excreted level of Cd and estimated glomerular filtration for both groups, and it was again significantly affected by other demographic factors. In the diabetic group it was negatively correlated with hypertension, which is not to be expected.

Author Response

Comments and Suggestions - REVIEWER 3

General comment. In general, there are five major remarks. First one, the methodology: three different methods for determining the Cd level used on different groups of subjects, significant disproportion in the number of subjects exposed to higher concentrations of Cd between diabetic and non-diabetic group, lack of the exact measurement of the plasma Cd concentrations, huge disproportion between the total number of non-diabetics and diabetics, differences between the groups in their demographic characteristics which have appeared to significantly affect multiple correlation analysis aiming to measure the strength of the dependence of risk from proteinuria and estimated glomerular filtration with the excreted concentrations of Cd. Due to these flaws the conclusion is overemphasized and compromised by the effect of those flaws on the overall results.

To retain the manuscript being further considered for publication it is mandatory that the authors recheck their exclusion and inclusion criteria, homogenize two groups of subjects, make them comparable in all parameters, even the extent of exposure to Cd and number of participants. After that has been done, they should rerun their statistical analyses and take into a consideration uncertainty factor occurring from the use of different methods in Cd concentration determination in different subjects.

General Response: We thank this reviewer for evaluating our manuscript. Below is point-by-point response to each comment.  Changes to the text are in blue.

  • Methodology
  • In the present study, the urinary Cd concentration was quantified by two methods, inductively coupled plasma mass spectrometry (ICP/MS) and atomic absorption spectrophotometry (AAS). The comparability of Cd quantities obtained was ascertained by simultaneous quantification of Cd in reference (standard) urine samples. We have inserted coefficient of variation statement to indicate the comparability of measured levels of Cd (lines 125-127).

  • Disproportionate number of subgroups
  • As this study is a population-based cross-sectional design, equal numbers in the subgroups are not necessary. To better reflect our work, the title of our paper has now been changed to “Tubular Dysfunction Is Significantly Worse in Type 2 Diabetics Exposed to Low Levels of Environmental Cadmium”.
  • Objective statements have been changed to read, “This study aimed to quantify Cd exposure, kidney tubular dysfunction and reduction of eGFR, in diabetics and non-diabetics who lived in low- and high-Cd exposure areas of Thailand”.
  • We first evaluated various associations by multiple regression analysis and logistic regression. We then estimated the effect size by univariate/covariance analysis that controlled for potential confounders.

  • Plasma Cd concentrations
  • The levels of Cd in plasma are in the nano molar ranges (nM). Thus, the quantification of plasma Cd required a highly sensitive analytical tool such as ICP//MS. However, the high cost of ICP/MS analysis and shipments of frozen blood and urine samples to Australia were prohibitive.

  • Influence of kidney function on proteinuria
  • This reviewer has correctly identified the need to adjust for kidney function to accurately evaluate an effect of Cd on the excretion rate of β2
  • To correct for differences in the number of surviving nephrons among study subjects, we normalized excretion of Cd and β2M to creatinine clearance (Ccr). We believe that Ccr-normalization which depicts an amount of a given chemical excreted per volume of filtrate, which is at least roughly related to the amount of the chemical excreted per nephron, is the best normalization method.

  • Overstated conclusion
  • Study objective statements have been changed, and part of the abstract has been rewritten to reflect our basic study design and revised conclusion as quoted below.
  • Conclusion: This study shows that tubular dysfunction and a reduced eGFR are more severe and more prevalent in diabetics than non-diabetics of similar age, BMI, and Cd body burden. Public health resources that promote a cessation of smoking and educate consumers about foods known to contain high levels of Cd are likely have significant health benefits.

Comment 1. Abstract

Line 28 – it is not clear how did the authors reach that conclusion. There is no mentioning of correlation analysis between Cd plasma content and clearance or proteinuria. The authors provide results of CDK biomarkers between subjects with and without diabetes, but, as they said, of the similar Cd clearance in both groups. How did the authors differentiate diabetes induced CDK and Cd induced CDK?

Response 1. Our manuscript has undergone extensive revisions to address the five major issues raised.  We provided here in a few examples of those changes.  Part of abstract has been changed as quoted below.

Abstract

“However, little is known about the nephrotoxicity of Cd in the diabetic population. Here, we compared Cd exposure, eGFR and tubular dysfunction in both diabetics (n = 81) and non-diabetics (n = 903) who were residents in low- and high-Cd exposure areas of Thailand”.

“A reduction in GFR and tubular dysfunction were more severe in diabetics than non-diabetics of similar age, BMI, and ranges of Cd body burden.  It thus appears that diabetics are more susceptible to Cd-induced nephropathy than non-diabetics.

We have provided further information regarding the environmental sources of Cd exposure among cohort participants in Section 2.1 (lines 89-97).

“As workplace exposure was an exclusion criterion, participants presumably acquired Cd from the diet and/or smoking. Based on the measured levels of Cd in duplicate diets [32] and a nation-wide survey of Cd levels in soils and food crops [33], environmental exposure to Cd in Bangkok and Pakpoon were low, and the Mae Sot district was high. From a previous study conducted in the Mae Sot district, the Cd content of the paddy soil samples exceeded the standard of 0.15 mg/kg, and rice samples collected from households contained 4-time the permissible Cd level of 0.1 mg/kg [34]. A health survey reported the prevalence of CKD in the Mae Sot as 6.1%, while the prevalence of tubular proteinuria was 36.1% [35].”

Comment 2. Materials and methods

Comments 2.1. Relevance of the Cd excretion level as the measure for Cd burdening of the organism is questionable in subjects with impaired or even diminished kidney function, which is the case in the diabetes subjects. As the authors mentioned, metallothionein chelated Cd is a measure of kidney damage. Thus, diabetes patients excreting chelated Cd and those excreting non-chelated one cannot be grouped together, and their results considered jointly in the comparison to the results of non-diabetic subjects. Further, it can not be deduced whether the CDK in those chelated Cd excreting patients is affected by diabetes or Cd, because there was not any comparison in the serum Cd levels between CDK-developed diabetics and diabetics with retained normal kidney function.

Response 2.1. Please see response to 1.4 above. To provide evidence that Cd influenced β2M excretion and/or GFR differently in diabetics and non-diabetics, the results of two additional multiple regression models have been added to Table 3 and 4. These additional analyses indicate that the effect of Cd on β2M was significant only in low body burden (ECd/Ccr <5 ng/L filtrate). As our study is a cross-sectional design, it is not possible to separate the effect of DM from that of Cd. We cautiously interpret these results to suggest that diabetics are more susceptible to the nephrotoxicity of Cd. Two recent histopathological studies using rats have been inserted (lines 361-363) to support this interpretation.

Comment 2.2. For three groups of subjects three different methods have been used to determine urine Cd concentration. The authors stated that each method was calibrated by the standard sample, but they do not mention whether or what is they ensured that Cd values measured by ICP-MS in a urine sample would give the same results if the same urine sample would be subject to AAS detection of Cd.

Response 2.2. Please see response to 1.4 above.

Comments 2.3. Diabetics and non-diabetics groups significantly differ in the number of subjects being exposed to low levels of Cd, being in favor of diabetics. This fact additionally undermines the conclusion provided by the authors regarding the Cd adverse effect on kidneys in diabetics. Besides, there is significant difference in the age, the group consisting of diabetic patients is older compared to the reference group. Furthermore, in the Table 2, Cd level is not identified as a risk factor affecting the prevalence odd ratio for proteinuria or estimated glomerular filtration.

Response 2.3.  Changes have been made to data in Table 2, where Cd has been shown to be inversely associated with eGFR in diabetic cases from a low-Cd exposure area.

Comment 2.4. Regarding the results presented in Figure 1, particularly 1b and 1d, it is difficult to draw any conclusions by comparing results for 11 or 4 diabetics with those for or 3 non-? diabetics, respectively. Again, regardless of the adjusted Cd excretion, in the group of diabetic subjects no difference in β2M excretion as a biomarker for proteinuria was recorded.

Multiple regression analysis showed a significant correlation between excreted Cd and proteinuria risk in the non-diabetic group. However, the result is significantly affected by other group characteristics, and which most interesting inverts the effect of blood pressure on β2M excretion, which is unusual.

Response 2.4.  With reference to Figures 1b and 1d, F and η2 values provided in these Figure are the comparison of non-DM vs. DM not subsets stratified by ECd/Ccr. We thank the Reviewer for raising this blood pressure effect. We will explore this phenomenon further in our future work. Although our data seemed to suggest a protective effect of high blood pressure (hypertension), most hypertensive subjects were treated. Thus, it was possible that it was an effect of anti-hypertensive medication.

We believe that a prospective case-control study design is required to address the mechanistic aspects. We discuss data from the Netherlands that unambiguously show the influence of smoking and Cd exposure on the onset and progression of diabetic kidney disease using a case-control study design (lines 386-392). Notably, however, eGFR was the only kidney parameter considered in the Netherlands studies [49,50]. In comparison, eGFR and tubulopathy both were investigated in our study.

49.Hagedoorn, I.J.M.; Gant, C.M.; Huizen, S.V.; Maatman, R.G.H.J.; Navis, G.; Bakker, S.J.L.; Laverman, G.D. Lifestyle-related exposure to cadmium and lead is associated with diabetic kidney disease. J. Clin. Med. 2020, 9, 2432.

  1. Oosterwijk, M.M.; Hagedoorn, I.J.M.; Maatman, R.G.H.J.; Bakker, S.J.L.; Navis, G.; Laverman, G.D. Cadmium, active smoking and renal function deterioration in patients with type 2 diabetes. Nephrol. Dial. Transplant 2022, gfac270.

Comments 2.5. Significant correlation between the excreted level of Cd and estimated glomerular filtration for both groups, and it was again significantly affected by other demographic factors. In the diabetic group it was negatively correlated with hypertension, which is not to be expected.

Response 2.5. Please see response 2.4 above.

Round 2

Reviewer 2 Report

The revised manuscript provides better content to understand this topic.

Author Response

Comments and Suggestions: REVIEWER 2

The revised manuscript provides better content to understand this topic.

We thank the reviewer for reviewing a revised version of our paper, and has found that it has been satisfactorily improved.

Reviewer 3 Report

The authors have implemented some recommendations. However, for results to be relevant it is mandatory to reorganize the study groups and to reduce significant difference in the number of subject originating from low-exposure locality between diabetic and nondiabetic group and than to rerun the analysis. It cannot remain 86.4% vs. 42.3%. As high difference significantly affects the results.

Author Response

Comments and Suggestions-REVIEWER 3

The authors have implemented some recommendations. However, for results to be relevant it is mandatory to reorganize the study groups and to reduce significant difference in the number of subject originating from low-exposure locality between diabetic and nondiabetic group and then to rerun the analysis. It cannot remain 86.4% vs. 42.3%. As high difference significantly affects the results.

Response: We thank the reviewer for providing further feedback and suggestions.  Accordingly, we have made the following changes to our manuscript.

  • We undertook resampling of the non-diabetic group such that it can now be considered as representative of the low Cd-exposure conditions.
  • After resampling the total number of participants was 674, and the overall percentage of those drawn from low-Cd exposure locations is increased from 45.9% to 67.1%.
  • The percentages of residents in low-Cd exposure locations in the non-diabetic group is increased from 42.3% to 64.4%.
  • Changes were also made to abstract, tthe data in tables 1-4, and figures 1-3 were reconstructed.
  • Changes to the text throughout the manuscript were also made.
  • Changes to the text are in blue.

We have maintained the overall conclusion (Section 5) because it is now justified by results obtained from resampling and reanalysis of the non-diabetic group which is representative of low environmental exposure to Cd.

Abstract has been rewritten and quoted below. 

The global prevalence of diabetes, and its major complication, diabetic nephropathy, have reached epidemic proportions. The toxic metal cadmium (Cd) also induces nephropathy, indicated by a sustained reduction in the estimated glomerular filtration rate (eGFR) and excretion of β2-microglobulin (β2M) above 300 µg/day, which reflects kidney tubular dysfunction. However, little is known about the nephrotoxicity of Cd in the diabetic population. Here, we compared Cd exposure, eGFR and tubular dysfunction in both diabetics (n = 81) and non-diabetics (n = 593) who were residents in low- and high-Cd exposure areas of Thailand. We normalized Cd and β2M excretion rates (ECd and Eβ2M) to creatinine clearance (Ccr) as ECd/Ccr and Eβ2M/Ccr. Tubular dysfunction and a reduced eGFR were respectively 8.7-fold (p < 0.001) and 3-fold (p = 0.012) more prevalent in the diabetic than the non-diabetic groups. Doubling of ECd/Ccr increased the prevalence odds ratios for a reduced eGFR and tubular dysfunction by 50% (p < 0.001) and 15% (p = 0.002) respectively. In a regression model analysis of diabetics from the low-exposure locality, Eβ2M/Ccr was associated with ECd/Ccr (β = 0.375, p = 0.001) and obesity (β = 0.273, p = 0.015).  In the non-diabetic group, Eβ2M/Ccr was associated with age (β = 0.458, p <0.001) and ECd/Ccr (β = 0.269, p <0.001). However, after adjustment for age, and body mass index (BMI), the Eβ2M/Ccr was higher in the diabetics than non-diabetics of similar ECd/Ccr ranges. Thus, tubular dysfunction was more severe in diabetics than non-diabetics of similar age, BMI, and Cd body burden.

Round 3

Reviewer 3 Report

The authors have implemented suggested revisions and the quality of the manuscript and relevance of the results are significantly increased.